# A Machine Learning-Driven Virtual Biopsy System For Kidney Transplant Patients

Daniel Yoo[1,25], Gillian Divard[1,2,25], Marc Raynaud[1], Aaron Cohen[3], Tom D. Mone[3], John Thomas Rosenthal[4], Andrew J. Bentall [5], Mark D. Stegall[6], Maarten Naesens[7], Huanxi Zhang[8], Changxi Wang[8], Juliette Gueguen[9], Nassim Kamar[10], Antoine Bouquegneau[11], Ibrahim Batal [12], Shana M. Coley[12], John S. Gill[13], Federico Oppenheimer[14], Erika De Sousa-Amorim[14], Dirk R. J. Kuypers[7], Antoine Durrbach[15], Daniel Seron[16], Marion Rabant[17], Jean-Paul Duong Van Huyen[1,17], Patricia Campbell[18], Soroush Shojai [18], Michael Mengel [18], Oriol Bestard[16], Nikolina Basic-Jukic[19], Ivana Jurić[19], Peter Boor [20], Lynn D. Cornell[21], Mariam P. Alexander [21], P. Toby Coates [22], Christophe Legendre[1,23], Peter P. Reese[1,24], Carmen Lefaucheur[1,2], Olivier Aubert[1,23] & Alexandre Loupy[1,23] ✉

In kidney transplantation, day-zero biopsies are used to assess organ quality and discriminate between donor-inherited lesions and those acquired post-transplantation. However, many centers do not perform such biopsies since they are invasive, costly and may delay the transplant procedure. We aim to generate a non-invasive virtual biopsy system using routinely collected donor parameters. Using 14,032 day-zero kidney biopsies from 17 international centers, we develop a virtual biopsy system. 11 basic donor parameters are used to predict four Banff kidney lesions: arteriosclerosis, arteriolar hyalinosis, interstitial fibrosis and tubular atrophy, and the percentage of renal sclerotic glomeruli. Six machine learning models are aggregated into an ensemble model. The virtual biopsy system shows good performance in the internal and external validation sets. We confirm the generalizability of the system in various scenarios. This system could assist physicians in assessing organ quality, optimizing allograft allocation together with discriminating between donor derived and acquired lesions post-transplantation.

In medicine, biopsy has become a standard test for establishing a diagnosis for both malignant and benign tumors as well as characterizing inflammatory diseases and other pathologic processes, thereby guiding therapeutic management[1].

In transplant medicine, the biopsy of the organ has been performed since the first pioneering work of Barry et al. and of Hamburger in Paris, becoming the gold standard for diagnosing allograft rejection and other various pathological processes that harm the allograft[2,3]. The histological evaluation of donors, also called "day-zero biopsies," has been implemented in several transplant programs[4–6] to judge the quality of a donor organ and, on occasion, to rule out the possibility of underlying diseases in donors[7]. In addition, day-zero biopsies provide a valuable baseline to which the findings of subsequent biopsies of the kidney allograft can be compared and may also advocate therapeutic strategies[8,9].

Despite their potential usefulness, day-zero biopsies are still not performed at many transplant centers and happen only in specific situations[10,11] since they remain invasive, time-consuming, and costly procedures that require organization of surgical, medical, pathological, and technical resources and might increase cold ischemia time associated with worst outcomes[12]. In addition, as we previously reported,

the organ quality assessment has become ever more important in the current worldwide increase of transplantation from older donors, donation after circulatory death, and donors with significant clinical risk factors to optimize the use of these kidneys to improve transplant outcomes[13–15]. These vulnerable organs may carry, at the time of transplantation, arteriosclerosis, fibrosis, hyalinosis, and glomerulosclerosis lesions[16]. If identified in a post-transplantation biopsy without the finding of a day-zero biopsy, these histological lesions, because of their non-specificity, might be wrongly attributed to calcineurin inhibitor toxicity, infectious diseases, or allo-immune response with significant impact for decision-making and patient management[6–8].

To circumvent these limitations, we designed a study to develop and validate a non-invasive virtual biopsy system that uses routinely collected donor parameters to predict the kidney day-zero biopsy findings to help physicians in guiding diagnostics, therapeutics, and immediate patient management post-transplant. The virtual biopsy system, an artificial intelligence model, provides virtual results that would have been obtained if a biopsy would have been performed. Since machine learning has demonstrated its clinical relevance in some medical specialities and comparative discriminative performance to logistic regression[17–20], we based our analyses on machine learning methods, using a large and qualified international cohort of donors who underwent routine and protocolized collection of donor parameters, together with day-zero biopsy assessment using the standards of the international Banff allograft histopathology classification[21].

## Results

### Baseline characteristics of the derivation cohort
We included a total of 12,402 day-zero biopsies from the 15-participating transplant centers for the derivation cohort. The mean donor age was $46.7 \pm 14.9$ (standard deviation, SD) years; 5450 (44.0%) were female, and 9395 (75.8%) were deceased donors. The mean serum creatinine was $1.2 \pm 1.0$ mg/dL. Baseline characteristics of the derivation cohort by country are shown in Table 1. The population is described in detail in Supplementary Method 1. Baseline characteristics of the derivation cohort stratified by center are described in Supplementary Table 1.

### Kidney histology lesions in the derivation cohort
Table 1 depicts the day-zero kidney biopsy findings of the derivation cohort. The median percentage of glomerulosclerosis was of 3.0% (interquartile range, IQR 0.0–10.0). The arteriosclerosis (Banff score cv) lesion score's distribution was 60.2%, 26.4%, 11.3%, and 2.1% for Banff scores None (Banff score 0), Mild (Banff score 1), Moderate (Banff score 2), and Severe (Banff score 3), respectively. The arteriolar hyalinosis (Banff score ah) lesion score's distribution was 68.8%, 21.3%, 8.1%, and 1.8% for scores 0, 1, 2, and 3, respectively. Finally, the interstitial fibrosis and tubular atrophy (Banff score IFTA) lesion score's distribution was 64.4%, 30.0%, 4.6%, and 1.0% for scores 0, 1, 2, and 3, respectively. Most moderate or severe (score 2 or 3) lesions of cv, ah, and IFTA were from deceased donors (Supplementary Table 2).

### Kidney virtual biopsy system development
The population cohort was imputed separately by derivation and external cohorts then pre-processed (Supplementary Tables 3, 4). We tuned and generated the best performing models for predicting the lesion scores, based on the donor parameters. The details of the hyperparameters tuning are available in Supplementary Table 5. Then, the ensemble model that groups these models together was generated. For each biopsy lesion score, we selected the ensemble models as a virtual biopsy system (see methods).

### Donor parameters' relative importance on lesion prediction
We examined the importance of the 11 donor parameters used for the virtual biopsy system development by averaging the importance produced by the models (Fig. 1). Overall, the three most important and predictive parameters for the biopsy lesions were age, serum creatinine, and the body mass index (BMI). The hypertension and cerebrovascular cause of death were the following highly important parameters overall.

### Model prediction performance on derivation cohort
The ensemble models showed discrimination performance during cross-validation with the multi-area under the curves (multi-AUC) of 0.833 (SD 0.013), 0.773 (0.020), 0.830 (0.027) for cv, ah, and IFTA lesions, respectively. Additionally, the ensemble models achieved area under the receiver operating characteristic curves (AUROC) of 0.880 (0.016), 0.823 (0.019), and 0.900 (0.023) for cv, ah, and IFTA lesions, respectively (Fig. 2). Ensemble models' cut-offs were calibrated to maximize Youden's J statistic. With the calibrated cut-offs of 0.582 for cv, 0.596 for ah, and 0.637 for IFTA, balanced accuracies (mean of sensitivity and specificity) were 0.786 (0.021) for cv, 0.736 (0.021) for ah, and 0.813 (0.024) for IFTA. For the glomerulosclerosis lesion, the mean absolute error (MAE) was 5.999 (0.032) and the root mean square error (RMSE) was 8.888 (0.059). The ensemble models and random forest models showed comparative performance. Table 2 summarizes the performances of all generated models. The detail cross-validation results are available in Supplementary Table 6. Calibration is shown as confusion matrix for each model in Supplementary Table 7.

### External validation of the virtual biopsy system
We included a total of 1630 day-zero biopsies from the USA and China for the external validation (Supplementary Method 1). Comparison with the derivation cohort and the baseline donor characteristics are available in Supplementary Tables 8, 9. The median percentage of glomerulosclerosis was 2.1% (IQR 0.0–12.5). The cv lesion score's distribution was 27.9%, 33.9%, 36.3%, and 1.9% for Banff scores None (Banff score 0), Mild (Banff score 1), Moderate (Banff score 2), and Severe (Banff score 3), respectively. The ah lesion scores 0 to 3 were distributed into 53.8%, 38.4%, 6.4%, and 1.4% for scores, respectively. The IFTA scores 0 to 3 were distributed into 40.4%, 30.7%, 28.7%, and 0.2%, respectively. Similar to the derivation cohort, most moderate or severe (score 2 or 3) lesions of cv, ah, and IFTA were from deceased donors (Supplementary Table 10).

In the Columbia University cohort, the ensemble models performed with the multi-AUCs of 0.740 (95% confidence interval [CI] 0.711–0.768), 0.733 (0.694–0.778), and 0.723 (0.705–0.772), for cv, ah, and IFTA lesions, respectively. Additionally, the ensemble models performed with the AUROCs of 0.880 (0.862–0.896), 0.922 (0.882–0.955), and 0.905 (0.889–0.920) for cv, ah, and IFTA lesions, respectively. With the same cut-offs obtained from internal validation, the balanced accuracies (mean of sensitivity and specificity) were 0.787 (0.764–0.808), 0.808 (0.741–0.872), and 0.843 (0.824–0.862) for cv, ah, and IFTA, respectively. For glomerulosclerosis, the ensemble model showed the MAE of 5.200 (4.971–5.422) and the RMSE of 6.630 (6.339–6.908).

In the Sun Yat-sen University cohort, the ensemble models showed the multi-AUCs of 0.740 (95% CI 0.663–0.807), 0.736 (0.654–0.821), and 0.798 (0.731–0.839) for cv, ah, and IFTA lesions, respectively. Furthermore, the AUROCs from the ensemble models were 0.902 (0.783–0.978), 0.895 (0.825–0.950), 0.935 (0.867–0.985) for cv, ah, and IFTA lesions, respectively. The balanced accuracies (same cut-offs obtained from internal validation), were 0.760 (0.578–0.950), 0.840 (0.762–0.899), 0.797 (0.638–0.959) for cv, ah, and IFTA lesions, respectively. For glomerulosclerosis, the ensemble model showed the MAE of 4.608 (4.229–4.989) and the RMSE of 5.731 (5.269–6.197) for glomerulosclerosis.

Figure 2 summarizes the performance of the ensemble models. Calibration in the external validation cohorts is shown in Supplementary Table 11.

**Table 1 | Baseline donor characteristics of population in the derivation cohort**

| | Overall(n = 12,402) | France(n = 2594) | USA(n = 5744) | Canada(n = 1578) | Australia(n = 370) | Belgium(n = 864) | Spain(n = 799) | Croatia(n = 453) |
|---|---|---|---|---|---|---|---|---|
| Age (years), mean (SD) | 46.7 (14.9) | 52.0 (16.2) | 43.6 (13.7) | 42.0 (13.5) | 46.9 (14.4) | 46.2 (12.8) | 61.1 (11.2) | 47.8 (12.2) |
| Sex female, No. (%) | 5450 (44.0%) | 1072 (41.3%) | 2581 (44.9%) | 757 (48.0%) | 187 (51.9%) | 386 (44.7%) | 281 (35.2%) | 186 (41.1%) |
| Donor type | | | | | | | | |
| Deceased donor, No. (%) | 9395 (75.8%) | 2528 (97.5%) | 3402 (59.2%) | 1065 (67.5%) | 284 (76.8%) | 864 (100.0%) | 799 (100.0%) | 453 (100.0%) |
| Death from circulatory disease, No. (%)[a] | 1471 (15.7%) | 195 (7.7%) | 531 (15.8%) | 126 (11.8%) | 65 (23.0%) | 225 (26.0%) | 329 (41.2%) | 0 (0.0%) |
| Death from cerebrovascular disease, No. (%)[a] | 4001 (42.8%) | 1391 (55%) | 942 (28.0%) | 326 (30.6%) | 113 (40.5%) | 433 (50.1%) | 513 (64.2%) | 283 (62.5%) |
| Diabetes mellitus, No. (%) | 782 (7.4%) | 175 (6.9%) | 428 (8.5%) | 48 (3.6%) | 10 (2.7%) | 3 (3.3%) | 111 (14.2%) | 7 (1.5%) |
| Hypertension, No. (%) | 2375 (21.1%) | 613 (24.9%) | 916 (18.1%) | 145 (11.1%) | 47 (12.8%) | 122 (14.4%) | 407 (52.2%) | 125 (27.6%) |
| BMI (kg/m²), mean (SD) | 26.9 (5.5) | 25.2 (4.7) | 28.1 (6.0) | 26.4 (5.3) | 26.8 (5.7) | 25.3 (4.1) | 27.6 (5.0) | 26.3 (3.6) |
| HCV status, No. (%) | 233 (1.9%) | 34 (1.4%) | 180 (3.2%) | 16 (1.1%) | 0 (0.0%) | 0 (0.0%) | 3 (0.4%) | 0 (0.0%) |
| Creatinine (mg/dL), mean (SD) | 1.2 (1.0) | 1.0 (0.6) | 1.6 (1.3) | 1.0 (0.6) | 0.8 (0.3) | 0.8 (0.5) | 1.0 (0.5) | 0.9 (0.4) |
| Proteinuria, No. (%) | 1904 (20.7%) | 1101 (49.2%) | 317 (6.2%) | 266 (35.6%) | 3 (3.7%) | 42 (42.4%) | 83 (17.9%) | 92 (20.3%) |
| Number of Glomeruli, mean (SD) | 39.3 (33.5) | 21.5 (15.5) | 65.0 (41.4) | 32.9 (24.8) | 38.2 (17.4) | 27.6 (16.7) | N/A | 57.0 (33.9) |
| Arteriosclerosis (cv) Banff score, No. (%) | | | | | | | | |
| 0 | 7073 (60.2%) | 915 (36.9%) | 3697 (65.6%) | 1007 (73.9%) | 81 (51.3%) | 765 (88.5%) | 305 (38.3%) | 303 (66.9%) |
| 1 | 3105 (26.4%) | 818 (33.0%) | 1335 (23.7%) | 263 (19.3%) | 48 (30.4%) | 76 (8.8%) | 425 (53.3%) | 140 (30.9%) |
| 2 | 1325 (11.3%) | 645 (26.0%) | 482 (8.5%) | 89 (6.5%) | 18 (11.4%) | 22 (2.5%) | 63 (7.9%) | 6 (1.3%) |
| 3 | 252 (2.1%) | 103 (4.2%) | 125 (2.2%) | 4 (0.3%) | 11 (7.0%) | 1 (0.1%) | 4 (0.5%) | 4 (0.9%) |
| Arteriolar hyalinosis (ah) Banff score, No. (%) | | | | | | | | |
| 0 | 8242 (68.8%) | 1010 (39.8%) | 4857 (86.8%) | 831 (54.2%) | 280 (79.3%) | 640 (74.2%) | 375 (59.2%) | 249 (55.0%) |
| 1 | 2546 (21.3%) | 950 (37.4%) | 548 (9.8%) | 415 (27.1%) | 63 (17.8%) | 178 (20.6%) | 214 (33.8%) | 178 (39.3%) |
| 2 | 968 (8.1%) | 462 (18.2%) | 141 (2.5%) | 251 (16.4%) | 8 (2.3%) | 41 (4.8%) | 42 (6.6%) | 23 (5.1%) |
| 3 | 217 (1.8%) | 117 (4.6%) | 52 (0.9%) | 37 (2.4%) | 2 (0.6%) | 4 (0.5%) | 2 (0.3%) | 3 (0.7%) |
| Interstitial fibrosis and tubular atrophy (IFTA) Banff score, No. (%) | | | | | | | | |
| 0 | 7822 (64.4%) | 1594 (62.0%) | 4072 (71.9%) | 830 (58.1%) | 328 (88.6%) | 648 (75.0%) | 185 (23.2%) | 165 (36.4%) |
| 1 | 3647 (30.0%) | 806 (31.3%) | 1229 (21.7%) | 549 (38.4%) | 37 (10.0%) | 198 (22.9%) | 572 (71.6%) | 256 (56.5%) |
| 2 | 562 (4.6%) | 131 (5.1%) | 293 (5.2%) | 48 (3.4%) | 4 (1.1%) | 14 (1.6%) | 41 (5.1%) | 31 (6.8%) |
| 3 | 117 (1.0%) | 41 (1.6%) | 68 (1.2%) | 1 (0.1%) | 1 (0.3%) | 4 (0.5%) | 1 (0.1%) | 1 (0.2%) |
| Glomerulosclerosis, median (interquartile range) | 3.0 (0.0-10.0) | 5.9 (0.0-13.3) | 0.0 (0.0-6.0) | 4.8 (0.0-9.5) | 3.9 (0.0-9.1) | 0.0 (0.0-8.3) | 7.4 (3.1-7.4) | 3.3 (0.0-7.7) |

Proteinuria values were positive when dipstick greater than or equal to 1 or urine protein to creatinine ratio (UPCR, g/g) greater than or equal to 0.5 g/g.
BMI body mass index, HCV hepatitis C virus.
[a]Number and % were calculated among deceased donors.

a

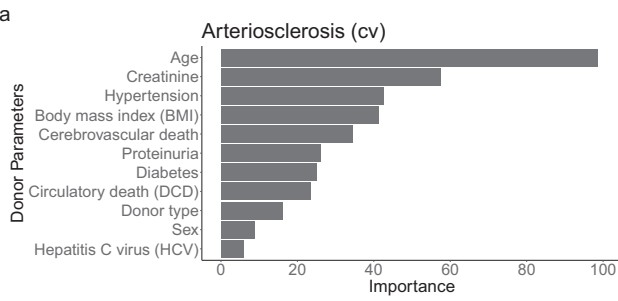

b

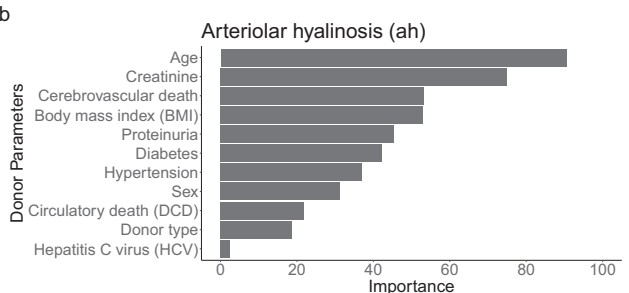

c

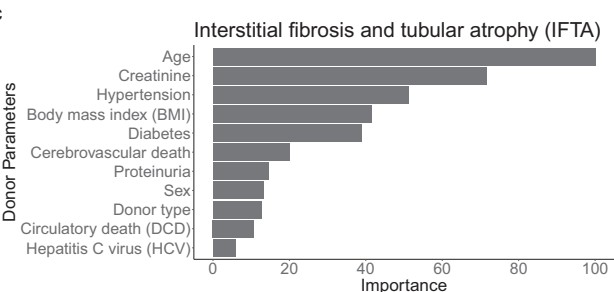

d

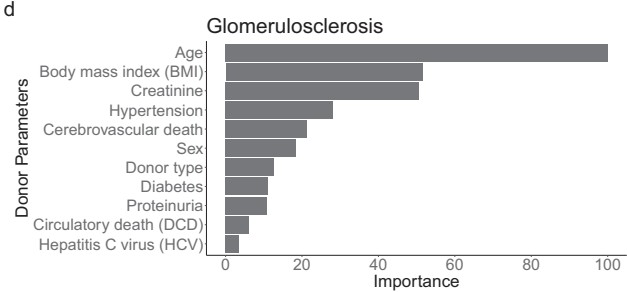

**Fig. 1 | Clinical and biological parameters' importance.** We performed random forest, gradient boosting machine, extreme gradient boosting tree, linear discriminant analysis, model averaged neural network, and multinomial logistic regression to measure the parameter importance for predicting the day-zero biopsy histological lesion scores during the derivation process. The importance was then averaged for the ensemble model. **a** Donor parameter importance for arteriosclerosis (cv Banff score). **b** Donor parameter importance for arteriolar hyalinosis (ah Banff score). **c** Donor parameter importance for interstitial fibrosis and tubular atrophy (IFTA Banff score). **d** Donor parameter importance for the percentage of sclerotic glomeruli (glomerulosclerosis score). Banff scores: cv arteriosclerosis, ah arteriolar hyalinosis, IFTA interstitial fibrosis and tubular atrophy. BMI body mass index, DCD donation after circulatory death, HCV hepatitis C virus. Source data are provided as a Source Data file.

### Validation of the virtual biopsy system in various scenarios
We confirmed the robustness of the virtual biopsy system in different subpopulations and clinical scenarios in the internal cross-validation, including (i) region (Europe, North America or Australia), (ii) donor ethnicity (African American, Caucasian, and Others [Hispanic, Asian, and Arabic]), (iii) donor criteria (extended criteria donors or standard criteria donors plus living donors), and (iv) biopsy type (pre-implantation and postreperfusion). Overall, the system showed good performance in subpopulations. These analyses are depicted in Supplementary Table 12.

### Pathologists' biopsy findings reliability
We confirmed the inter-pathologist consistency in four expert nephropathologists from Necker hospital and Mayo clinic in evaluating the biopsy findings, with Fleiss Kappas of 0.68 (95% CI 0.63–0.73), 0.59 (0.53–0.65) and 0.51 (0.44–0.59), for cv, ah, and IFTA lesions respectively. The overall Fleiss Kappa for all lesions was 0.63 (0.60–0.66).

### Performance of kidney donor profile index (KDPI) score
The derivation cohort included 4241 biopsies, and the external validation cohort comprised 1124 biopsies (920 from Columbia University medical center and 204 from Sun Yat-sen University). The mean KDPI was 53.43 (SD 29.49) in the derivation cohort and 63.24 (SD 26.63) in the external validation cohort.

Supplementary Table 13 shows model performance with KDPI as a parameter. The KDPI-based model achieved multi-AUCs of 0.688, 0.644, and 0.716 for cv, ah, and IFTA lesions during internal validation, respectively. Predicting glomerulosclerosis performed with the MAE of 6.647. During external validations, the KDPI-based model showed predictive performance for cv, ah, and IFTA, achieving multi-AUCs of 0.625, 0.668, and 0.638 for the Columbia University cohort, and 0.659, 0.552, and 0.710 for the Sun Yat-sen University cohort, respectively.

### Virtual biopsy system online application for physicians
Based on these results, we constructed a ready-to-use online application to offer physicians an open access to the virtual day-zero biopsy system (Supplementary Movie 1). The application allows physicians to enter a single patient's data, to get (i) the personalized probabilities of belonging to each day-zero histological lesion score and (ii) the prediction visualization with radar chart. The application is available online: https://transplant-prediction-system.shinyapps.io/Virtual_Biopsy_System. Figure 3 and Supplementary Fig. 1 provide examples of usage of the application in clinical practice with real donor clinical cases depicted. The potential clinical utility and impact of this application is also depicted in Supplementary Fig. 2.

## Discussion
In this international, multicohort study of kidney transplant biopsies from 17 worldwide centers including the largest Organ Procurement Organization (OPO) in the USA and labeled by expert kidney pathologists, we derived and validated a virtual biopsy system that uses non-invasive and routinely collected donor parameters to predict kidney histological lesions. The virtual biopsy system was developed with four ensemble models based on aggregation of six machine learning algorithms to decrease the bias and maximize the generalizability and predict four biopsy lesion results. Overall, the virtual biopsy system showed good discrimination, calibration, robustness, and generalizability in various countries, external validation cohorts, and clinical scenarios.

Over the past decade, the use of kidneys from older donors with comorbidities has expanded the pool of kidneys, raising the question of whether pathological examination of donated kidneys could help better characterize organ quality or drive inefficiencies in organ allocation[22]. Additionally, this biopsy procedure needs to be performed and interpreted by trained experts, which is difficult to implement 24/7[23]. Furthermore, in the USA, the United Network for Organ Sharing policy for organ allocation, recommends the use of

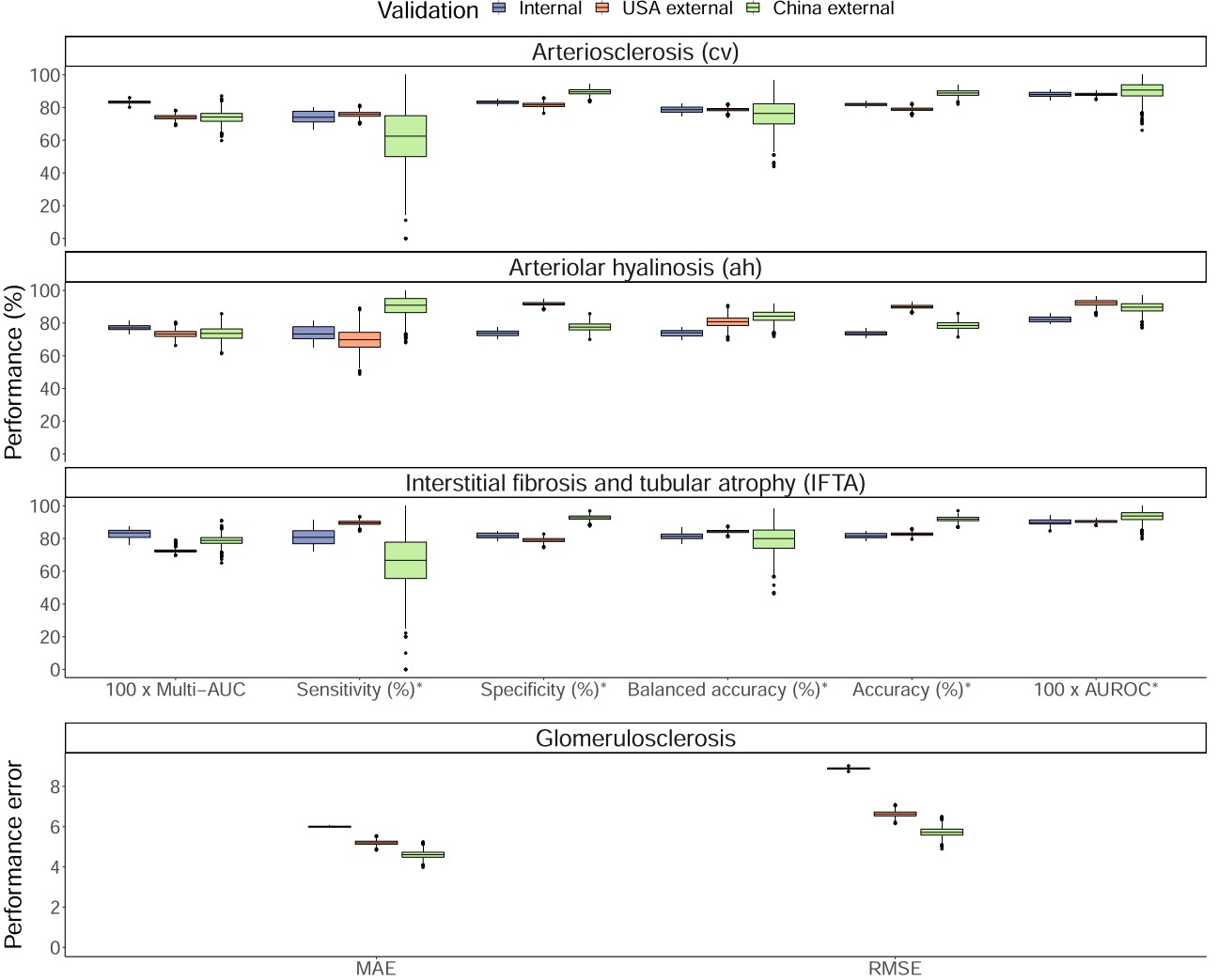

**Fig. 2 | Performance metrics of ensemble models across internal and external validation cohorts.** Ensemble models were internally and externally validated on the 3-times repeated 10-folds cross-validation and the external validation cohorts comprising Columbia university from the USA and Sun Yat-sen university from China. For multi-AUC, the full lesion scores were used. For other metrics, such as AUROC and sensitivity, categorical Banff scores (arteriosclerosis [cv Banff score], arteriolar hyalinosis [ah Banff score], and interstitial fibrosis and tubular atrophy [IFTA Banff score]) were dichotomized. Cut-offs were calibrated based on internal validation (cross-validation): 0.582, 0.596, 0.637 for cv, ah, IFTA lesions, respectively. For internal validations, performance was assessed in 30 resamples during cross-validation. For external validations, performance was assessed using 1,000 times bootstrapping. All box plots comprise the median line, the box indicated the interquartile range (IQR), whiskers denote the rest of the data distribution and outliers are denoted by points greater than ±1.5 × IQR. * For sensitivity, specificity, balanced accuracy, accuracy, and AUROC, the Banff lesion scores, cv, ah, and IFTA were dichotomized (scores 0–1 as negative and 2-3 as positive). Banff scores: cv arteriosclerosis, ah arteriolar hyalinosis, IFTA interstitial fibrosis and tubular atrophy. multi-AUC multi-area under the receiver operating characteristic curve, AUROC area under the receiver operating characteristic curve, MAE mean absolute error, RMSE root mean square error. Source data are provided as a Source Data file.

KDPI, day-zero biopsy results, and donor characteristics to assess organ quality before transplantation. Despite the importance, the lost time due to this procedure could be precious when the biopsy result is used for allocation purposes as every additional hour of cold ischemia time is highly associated with worse graft outcomes. Therefore, many centers are discouraged from performing day-zero biopsy because it remains an invasive and time-consuming procedure that could increase cold ischemia time[10,11].

Our literature search (Supplementary Method 2) revealed a dearth of studies that address the creation of a virtual biopsy for evaluating biopsy lesion presence and severity by utilizing non-intrusive factors such as donor parameters. Meanwhile, non-invasive diagnosis using machine learning has been studied. Yin et al. demonstrated that the potential of multiple machine learning classifiers in distinguishing histological features in bladder tumor images[24]. Detecting kidney biopsy results has been explored predominantly with histological

images using deep learning. In 2018, Marsh et al. developed a convolutional neural networks model to identify and classify glomerulosclerosis in day-zero kidney biopsies, improving pre-transplant evaluation[25]. Hara et al. showcased a U-Net based segmentation model for classifying normal and abnormal tubules in kidney biopsies[26]. However, a need persists to compensate for the absence of performed day-zero biopsy for kidney allografts by virtually assessing the presence and severity of biopsy lesions using non-invasive donor parameters.

In this context, we believe that the virtual biopsy system has many potential implications. First, it not only predicts the presence of lesions (binary classification) but also predicts the severity grades of the lesion (multiclass classification), which fosters a more complete clinical interpretation.

Second, the virtual biopsy system can help a physician to evaluate and contextualize post-transplant lesions, which might be inherited

**Table 2 | Machine learning classifiers' and ensemble models' performances**

| Models | Hand and Till'sMulti-AUC | | | Mean Absolute Error |
| --- | --- | --- | --- | --- |
| | Arteriosclerosis(cv Banff score) | Arteriolar hyalinosis(ah Banff score) | Interstitial fibrosis tubular atrophy(IFTA Banff score) | Glomerulosclerosis in percentage |
| Random Forest | 0.836 | 0.774 | 0.830 | 5.807 |
| Gradient Boosting Machine | 0.807 | 0.750 | 0.805 | 6.486 |
| Extreme Gradient Boosting Tree | 0.830 | 0.767 | 0.827 | 5.768 |
| Linear Discriminant Analysis[a] | 0.761 | 0.703 | 0.750 | -[a] |
| Model Averaged Neural Network | 0.777 | 0.720 | 0.757 | 6.573 |
| Multinomial Logistic Regression[a] | 0.763 | 0.706 | 0.753 | -[a] |
| Ensemble Model | 0.833 | 0.773 | 0.830 | 5.999 |

The models used for ordinal scores (multiclass classification) are as follows: random forest, gradient boosting machine, extreme gradient boosting tree, linear discriminant analysis, model averaged neural network, and multinomial logistic regression. The models used for the percentage of glomerulosclerosis (regression) are as follows: random forest, gradient boosting machine, extreme gradient boosting tree, and model averaged neural network. Finally, we created ensemble models; for the ordinal day-zero lesion scores, we averaged the probabilities of the six models; for the percentage of glomerulosclerosis, we used linear regression of the four models we created. For the ordinal day-zero lesion scores, model performances were assessed by Hand and Till's area under the curve (multi-AUC). For the percentage of glomerulosclerosis, model performances were assessed by mean absolute error (MAE). Ensemble models were selected as virtual biopsy system. Model performances were assessed in 3-times repeated 10-folds cross-validation (30 resamples).

*AUC area under the curve (higher the better), MAE mean absolute error (lower the better).*

[a]Linear discriminant analysis and multinomial logistic regression are not developed for regression but for classification.

from the donor or acquired after transplantation; this could reinforce precision medicine and patient monitoring of these nonspecific histological lesion to guide therapeutics[27–30].

Third, since the virtual biopsy system is trained on high-quality data and biopsies labeled by expert kidney pathologists, its inferences are highly reliable. Because the day-zero biopsy labeling depends on the skills and experience of observers (e.g., general or kidney pathologist) and temporal settings, the virtual biopsy system may partly address the current issues of Banff classification using histology such as physicians' variability and reproducibility in labeling biopsy findings. Additionally, it may have a great interest in many centers, especially from developing countries, that currently cannot yet afford to perform neither digital pathology with whole-slide imaging, nor day-zero biopsies due to the lack of resources.

Fourth, the system could decrease cold ischemia time and mobilization of team resources by circumventing the standard of care day-zero biopsy procedure using basic donor characteristics and virtual biopsy. Eliminating the process by offering the virtual biopsy could shorten the allocation time and improve the graft outcomes[31]. Overall, this can be achieved by utilizing the virtual biopsy system before organ retrieval (procurement) to provide physicians with a reliable surrogate of the true day-zero biopsy.

Fifth, the virtual biopsy system may be attractive for clinical trials by helping to improve the randomization of the patients at the time of transplantation, using not only the baseline characteristics but also the chronic lesions of kidney donors to avoid selection bias. Moreover, the efficacy of a new treatment is very often based on protocol biopsies where chronic lesions such as fibrosis and arteriosclerosis can be found. Because antibody-mediated rejection or immunosuppressive toxicity can induce those lesions[27–29], knowing their origin—whether they were inherited from the donor or from the consequence of treatment inefficacy—is crucial to avoid misinterpretation of the findings and loss of potential useful treatments[6,8].

Last, although the rapid improvements in computing power and huge digitized medical history records have led many researchers to attempt integrative approaches to scrutinizing unknown fields of medicine[17,32,33], it is still difficult for health professionals to approach these tools in real life. Since the virtual biopsy system is not a mere proof of concept, we generated an easy-to-use online application to support physicians and reinforce applicability. This online clinical application is available immediately. Beyond transplantation, the idea of a virtual biopsy system, using routinely accessible donor parameters to predict biopsy findings with the power of algorithms, can be easily cross-fertilized with other fields of medicine that have a comparable need to predict specific lesions for an enhanced interpretation of patient prognosis.

Our study has numerous strengths, but we also acknowledge the following limitations.

First, due to the multi-centre nature of the study, the problem of interobserver variability in labeling biopsy findings, practices, and procedures may have carried compatibility issues and impacted the study results[23]. However, we made four pathologists reassess 10% of the biopsies and showed that this variability was limited, confirming that the biopsy findings may be considered reliable. Besides, our data collection procedure followed high-quality structured protocols to ensure compatibility across study centers. Second, due to the large number of centers involved, some heterogeneity was induced in biopsy techniques, tissue processing, and tissue stain but reflecting the different practices worldwide and remains a limited part of the derivation cohort <7% used only hematoxylin and eosin stain or only frozen tissue. This heterogeneity makes the model more generalizable and robust by improving its exportability. Third, wedge biopsy increases the risk of capturing only subcapsular tissue which could underestimate the extent of vascular intimal thickening or overestimate glomerulosclerosis; this could have introduced unintended

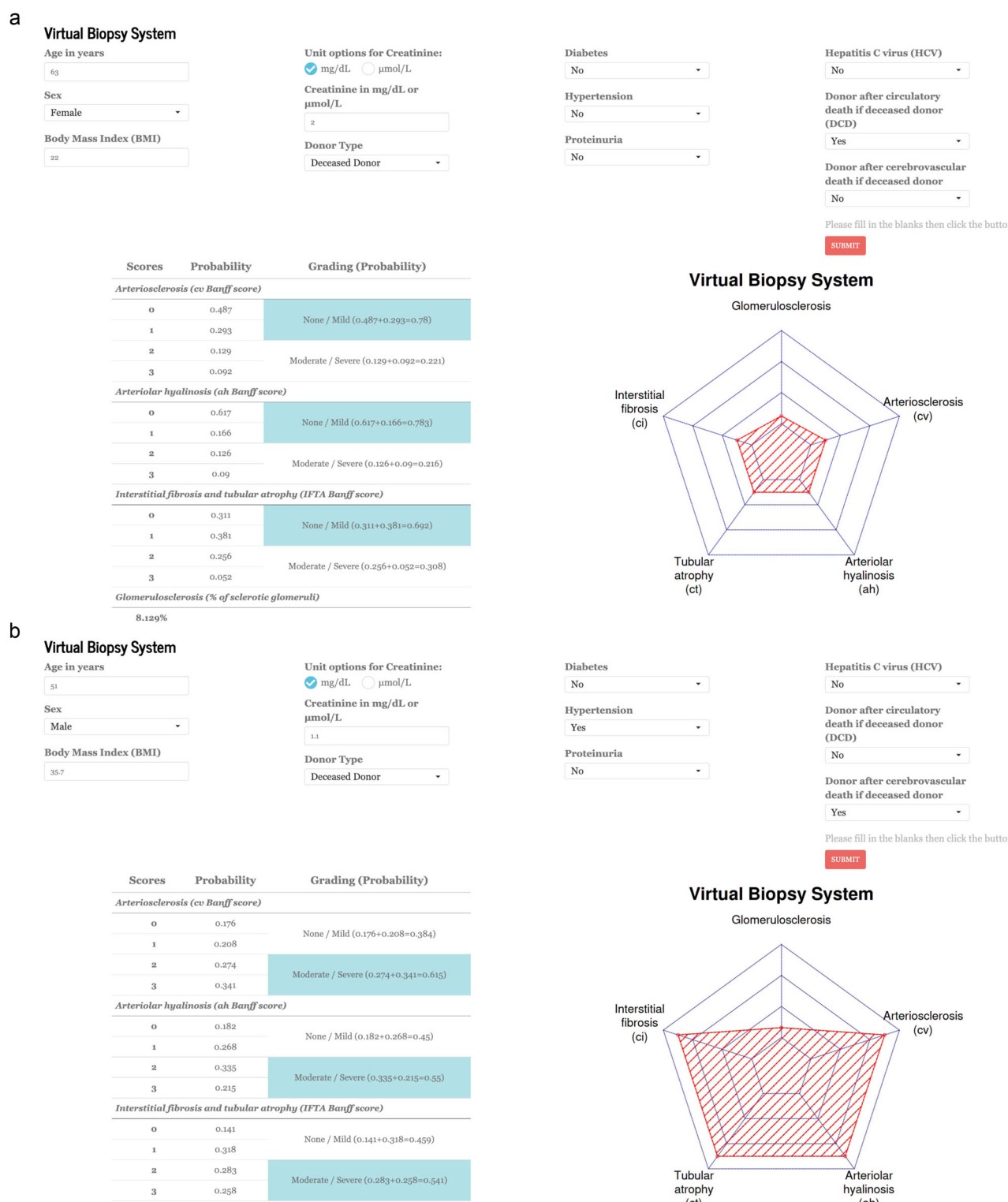

**Fig. 3 | Ready-to-use online application for physicians.** The online application aims to help physicians freely use the virtual day-zero biopsy findings for post-transplant patient management. **a** A virtual biopsy finding from 63-year-old female donor from circulatory cause of death with moderate BMI but poor kidney function (creatinine). **b** A virtual biopsy finding from 51-year-old male donor from cerebrovascular cause of death with high BMI and hypertension but moderate kidney function. Banff scores: cv arteriosclerosis, ah arteriolar hyalinosis, IFTA interstitial fibrosis and tubular atrophy, ci interstitial fibrosis, CT tubular atrophy, BMI body mass index, DCD donation after circulatory death, HCV hepatitis C virus Link to the app: https://transplant-prediction-system.shinyapps.io/Virtual_Biopsy_System.

overestimation in cv and glomerulosclerosis lesions[34,35]. However, most biopsies included were from centers that used core needle biopsy instead of wedge biopsy. Additionally, to overcome this issue, we included only centers with a large number of kidney transplants with a relatively low number of inadequate biopsies (7.2%) as compared to the literature (30%)[36]. Fourth, additional predictors, such as gene expression or new biomarkers, beyond the 11 donor parameters used to derive the virtual biopsy, may improve its performances. However, the parameters used in this study are the most commonly accessible, and including less standard ones might not only increase the number of missing data but also reduce generalizability by increasing the risk of parameters missing. Last, other sampling methods such as nested cross-validation, may help provide more precise prediction performances. However, with the large derivation cohort from heterogeneous and various data sources, we are confident in performing 3-times repeated 10-folds cross-validation for internal validation[37]. Moreover, we performed model assessments in subpopulations and various clinical scenarios. Finally, we showed the model performances are comparable in internal and external validations.

In conclusion, we derived and validated a machine learning-driven virtual kidney allograft biopsy system that uses easily accessible donor parameters at the time of transplantation. The virtual biopsy system demonstrates good performances and robustness across 17 geographically distinct centers and in many clinical scenarios. This system can provide physicians with a reliable estimation of the day-zero biopsy findings, which may reduce costs of invasive and time-consuming procedures and help guide further biopsy interpretations and patient management.

## Methods

### Study design and population

The population consisted of living or deceased and transplanted or discarded adult donors for kidney transplantation enrolled from January 1st, 2000, to December 31st, 2021, who underwent kidney biopsies performed prior to kidney transplantation as part of standard of care. For the derivation cohort, the study involved 15 centers including 14 institutions from seven countries (France, Belgium, Croatia, Spain, United States, Canada, and Australia) and the largest OPO in the USA (OneLegacy). For the external validation cohorts, two institutions from two countries were involved: Columbia university medical center from the USA and Sun-Yat-sen university from China. A total of 15,121 kidney biopsies were assessed overall. Exclusion criteria were inadequate biopsies according to Banff international classification requirements ($n = 1089$, 7.2%)[21]. A total of 14,032 kidney allograft biopsies were included for the final analyses including 1372 (9.8%) from discarded kidneys. Among them, 12,402 were in the derivation cohort and 1630 were in the external validation cohorts.

### Inclusion and ethics statement

All data were anonymized, and the clinical and biological data were collected from each center and entered into the Paris Transplant Group database (French data protection authority (CNIL) registration number 363505). On January 1st, 2021, the data were accessed from the database. On November 19th, 2021, the Chinese data were accessed from the database. On June 8th, 2022, the OneLegacy OPO data were accessed from the database. The protocol of this study (NCT04759209) was approved by the Paris Transplant Group's Institutional Review Board (IRB). Written informed consent was given by all living donors at the time of transplantation. The IRB of Paris Transplant Institute approved the study and waived the informed consent for deceased donors (registration no. 2018-1017-Virtual-Biopsy). The original collection and exportation of the data had the approval of the Ministry of Science and Technology for Sun-Yat-sen university in China. All data from the Paris Transplant Group centers (Necker, Saint Louis, and Toulouse Hospitals) were entered

prospectively at the time of transplantation; a structured protocol was used to ensure harmonization across study centers. To ensure data accuracy, an annual audit was performed. As part of standard clinical procedures, other datasets from the European, North American, Australian, and Asian centers were compiled, entered in the databases of the centers in accordance with local and national regulatory standards, and submitted to the Paris Transplant Group anonymously.

### Kidney biopsy histological assessment and protocols

Day-zero biopsies were performed after the organ was removed from the donor in accordance with standard practices by a surgeon using a 16-gauge needle device or a straight blade. The tissue was immediately fixed in an aqueous formaldehyde solution (formalin) or alcohol–formalin–acetic acid solution and subsequently embedded in paraffin or immediately frozen. The biopsy sections (4 µm) were stained with periodic acid-Schiff, Masson's trichrome, hematoxylin, and eosin. Using the international Banff classification kidney lesions scoring system[21], expert kidney pathologists graded the graft biopsy lesions using the following criteria: glomeruli number, arteriosclerosis, arteriolar hyalinosis, interstitial fibrosis and tubular atrophy, and the percentage of sclerotic glomeruli. A detailed table summarizing the participating centers' biopsy practices and procedures is presented in Supplementary Table 14.

### Outcomes of interest

The outcomes of interest were the biopsy findings according to the international Banff classification of allograft pathology, which uses a validated semi-quantitative ordinal grading scheme for all kidney compartments including: (i) arteriosclerosis defined by arterial intimal thickening in the most severely affected artery (Banff "cv" score), (ii) arteriolar hyalinosis defined by periodic acid-Schiff (PAS)-positive arteriolar hyaline thickening (Banff "ah" score), and (iii) interstitial fibrosis and tubular atrophy (Banff "IFTA" score) computed with the extent of cortical fibrosis (Banff "ci" score) and cortical tubular atrophy (Banff "ct" score)[21]. These semi-quantitative lesion grading scores are not linear. Last, the continuous percentage of sclerotic glomeruli was defined by the percentage of the total number of glomeruli affected by global sclerosis ("glomerulosclerosis" score)[5]. The Banff grading scheme in detail is available in Supplementary Method 3 and Supplementary Table 15.

### Candidate predictors of kidney biopsy histological lesions

Eleven candidates[2], universally collected donor predictors at donation, of kidney day-zero histological lesions were examined. They comprised donor's age, sex, type (living or deceased donor), donor's cerebrovascular cause of death, donor's circulatory cause of death (DCD), donor's history of hypertension, diabetes, hepatitis C virus (HCV) status, body mass index (BMI), lowest serum creatinine at donation, and donor proteinuria status. The details of these predictors are available in Supplementary Method 4.

### Statistical analyses

We used TRIPOD (Transparent Reporting of a Multivariable Prediction Model for Individual Prognosis or Diagnosis) statement for the reporting of the development and validation of the virtual biopsy system[38], which was adapted to machine learning (Supplementary Method 5). Figure 4 summarizes the process of generating and validating machine learning models.

### Descriptive analyses of baseline characteristics

For continuous variables, means and standard deviations or medians and interquartile ranges were used. We compared means and proportions between groups using Student's t-test, analysis of variance (ANOVA) (or Mann-Whitney test and Kruskal-Wallis if appropriate), or

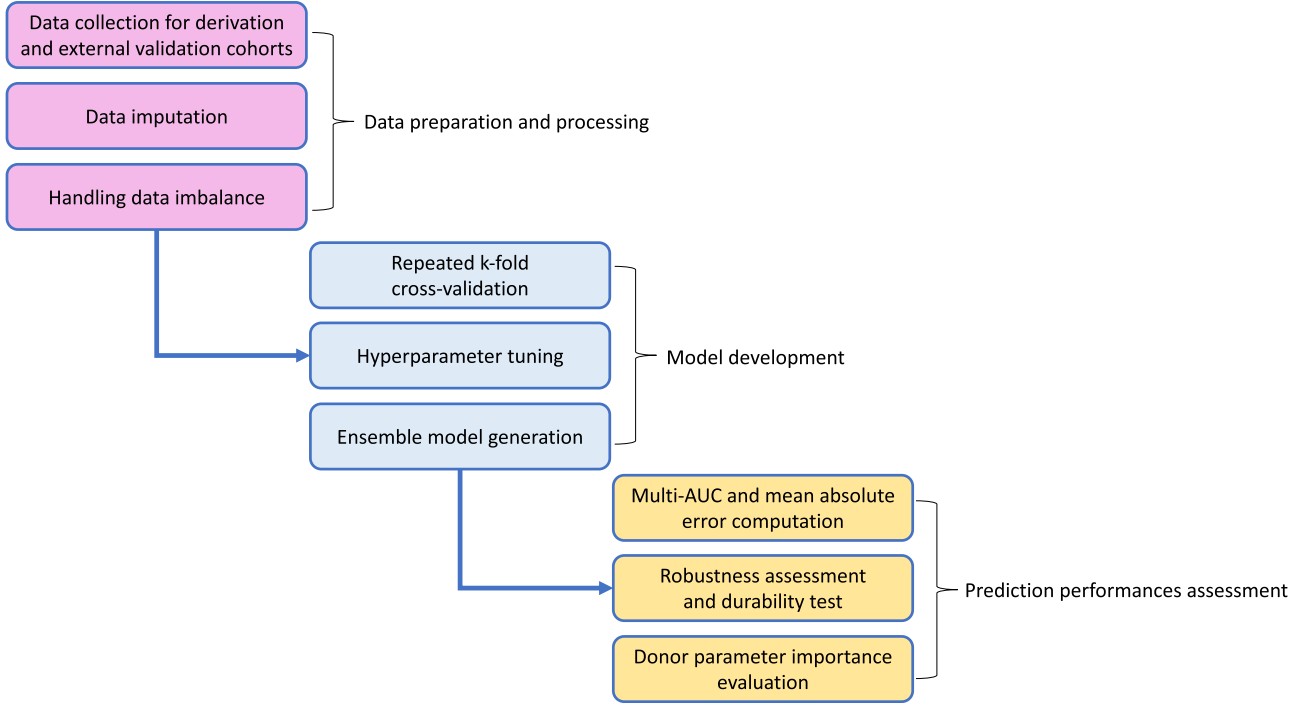

**Fig. 4 | Flow chart of virtual biopsy system machine learning pipeline.** The study comprises three main processes to develop and validate the virtual biopsy system for kidney transplant patients. Each step also comprises three sub-processes. multi-AUC multi-area under the receiver operating characteristic curve.

the chi-squared test (or Fisher's exact test if appropriate). All tests were two tailed.

## Algorithm pre-process

To minimize the data imbalance in the lesion scores and maximize the predictive performance, which had more mild/lower grades (over-represented) than severe/higher grades (under-represented), we applied an up-sampling method during the model training process by resampling random kidneys from the severe/higher grades. Three numeric continuous donor parameters (age, body mass index, and creatinine) were standardized to have mean of zero and a standard deviation of one. These pre-process steps were done with *caret* R Package[39].

## Development of the virtual biopsy system

To develop the virtual biopsy system, we computed probabilities for each day-zero histological lesion score from six machine learning models: random forests (RF)[40], model averaged neural networks (avNNet)[41], gradient boosting machine (GBM)[42], extreme gradient boosting tree (XGBoost)[43], linear discriminant analysis (LDA)[41], and multinomial logistic regression (MNOM)[44]. To avoid overfitting and sampling bias, hyperparameters were optimized by robust 3-times repeated 10-folds cross-validation when tuning the models[45]. Then, we aggregated the classification models by averaging probabilities provided by each model: this generated an ensemble model, or meta-classifier, which is aimed at decreasing bias and overfitting to take into account the "no free lunch" theorem[46–48]. MNOM and LDA were not used to predict glomerulosclerosis lesion (regression) because they are exclusively designed to predict categorical variables (classification). For the regression model, we built a linear model of regression models to create an ensemble model, a meta-regression.

## Virtual biopsy system prediction performances

Models' performances were assessed as internal and external validation. For the internal validation, the performance was assessed in 30 resamples from the 3-times repeated 10-folds cross-validation on

the derivation cohort. For the external validation, the performance was assessed on the external cohorts. To assess the discrimination performance of the machine learning models used for glomerulo-sclerosis, which is continuous, we used the MAE and RMSE as a supplementary metric[49]. For ordinal day-zero histological lesion scores, cv, ah, and IFTA, we used the multi-area under curve (multi-AUC) using Hand and Till's formula[50]. Further supplementary metrics for cv, ah, and IFTA were also reported for both internal and external validation: sensitivity, specificity, balanced accuracy (average of sensitivity and specificity), accuracy, and area under the receiver operating characteristic curve (AUROC). To present these supplementary metrics, we dichotomized the categorical Banff scores "None" (Banff score 0) and "Mild" (Banff score 1) as the negative class and "Moderate" (Banff score 2) and "Severe" (Banff score 3) as the positive class. Cut-offs for dichotomized Banff lesions were calculated using the method of Youden's J statistic on internal validation[51]. Supplementary Method 6 contains the rationale for the cut-offs used to measure the performance. The 1000 bootstraps were used to obtain 95% CIs while the external validation cohorts' samples were used for point estimate for each metric.

Model calibration was examined with confusion matrices. Furthermore, to assess the donor parameters that drive the performance of the models, we averaged the feature importance by RF, GBM, XGBoost, LDA (for classification models only), avNNet, and MNOM (for classification models only).

## Imputation of missing data

For biopsies with at least one missing data element for predictors of interest, random forest imputation algorithm was performed using the missForest R package[52]. The maximum iteration was set to 10 times for imputation. The details of the imputation process and results are presented in Supplementary Method 7.

## Kidney donor profile index (KDPI)

We conducted a sensitivity analysis to investigate whether KDPI could predict the day-zero biopsy lesions. We developed a model using only

the KDPI score. Biopsies from living donors and those with missing ethnicity, height, or weight data were excluded from the imputed dataset. Organ Procurement and Transplantation Network (OPTN) guidelines, based on the database as of April 07, 2023, were followed for KDPI calculations. An ensemble of RF, XGBoost, LDA, avNNet, and MNOM models was employed. LDA and MNOM were excluded for predicting glomerulosclerosis lesion. GBM was excluded due to the difficulty of deriving a univariate model.

## Assessment of the consistency in the biopsy evaluation

To evaluate the inter-pathologist's consistency in evaluating the biopsy findings, we randomly selected 10% of the biopsies and made them reassessed in the original two transplant centers (Necker Hospital and Mayo Clinic) by four expert nephropathologists. Pathologists were blinded to the previous biopsy findings. Fleiss Kappa was used to measure the consistency and was weighted to take into account the magnitude of errors in the re-assessment.

## Software and package

Descriptive analyses and machine learning analyses were conducted using R (version 3.5.1, R Foundation for Statistical Computing) and RStudio (version 2022.7.2.576). Packages used for data and machine learning analyses were: randomForest (version 4.6-14), gbm (version 2.1.5), xgboost (version 1.4.1.1), plyr (version 1.8.4), MASS (version 7.3-51.4), nnet (version 7.3-12), caret (version 6.0-84), caretEnsemble (version 2.0.1), tidyverse (version 1.3.0), ggsci (version 2.9), rsample (version 0.1.1), tidymodels (version 0.0.2), patchwork (version 1.0.0), dplyr (version 1.0.7), ggplot2 (version 3.3.1), yardstick (version 0.0.8), readr (version 1.3.1), cvms (version 1.3.3), pROC (version 1.18.0), rlist (version 0.4.6.2), autoxgboost (version 0.0.0.9000), shiny (version 1.6.0), shinythemes (version 1.1.2), kableExtra (version 1.3.4), and compareGroups (version 4.0.0).

## Reporting summary

Further information on research design is available in the Nature Portfolio Reporting Summary linked to this article.

# Data availability

The figure data generated in this study have been deposited in the public Synapse database (https://www.synapse.org/#!Synapse:syn51702348/files/)[53]. The figure data can be obtained by the signing-in process. The raw data are available from the corresponding author. Source data are provided with this paper.

# Code availability

Complete code to reproduce the figures is available in the synapse public Synapse database (https://www.synapse.org/#!Synapse:syn51702348/files/)[53]. A sign-in process is required to access the code.

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

## Acknowledgements

We thank Sophie Ferlicot, Sandra Cockfield, Sumit Mohan, Syed A. Husain, David J. Cohen, Lloyd E. Ratner, and Maisarah Jalalonmuhali for data acquisition. French government managed by the National Research Agency (ANR) with the grant agreement ANR-17-RHUS-0010 and European Union's Horizon 2020 research and innovation program EU-TRAIN with the grant agreement no. 754995 provided financial support. The funders of this study had no role in the study design, data collection, analysis, or interpretation of the manuscript.

## Author contributions

A.L. and O.A. supervised the study. D.Y., G.D., M.R., O.A., and A.L. designed the study, analyzed and interpreted the data, wrote and edited the manuscript. D.Y., G.D., M.R., A.C., T.M., J.R., A.J.B., M.D.S., M.N., H.Z., C.W., J.G., N.K., A.B., I.B., S.M.C., J.S.G., F.O., E.D.S.-A., D.R.J.K., A.D., D.S., M.R., J.-P.D.V.H., P.C., S.S., M.M., O.B., N.B.-J., I.J., P.B., L.D.C., M.P.A., P.T.C., C.L., P.P.R., C.L., O.A., and A.L. contributed to the data acquisition and D.Y., G.D., M.R., O.A., and A.L. verified the data. D.Y. performed the data analysis. D.Y., G.D., M.R., O.A., and A.L. wrote the manuscript. The corresponding author attests that all authors have read and approved the manuscript. A.L. was responsible for the decision to submit the manuscript for publication. All authors revised the manuscript.

## Competing interests

A.L. holds shares in Predict4Health, a software company that is not involved in the present research. The other authors declare no competing interests.

## Additional information

[1]Université Paris Cité, INSERM U970 PARCC, Paris Institute for Transplantation and Organ Regeneration, F-75015 Paris, France. [2]Kidney Transplant Department, Saint-Louis Hospital, Assistance Publique – Hôpitaux de Paris, Paris, France. [3]OneLegacy, Los Angeles, CA, USA. [4]David Geffen School of Medicine at UCLA, Los Angeles, CA, USA. [5]Division of Nephrology and Hypertension, Mayo Clinic Transplant Center, Rochester, MN, USA. [6]Department of Surgery, Mayo Clinic, Rochester, MN, USA. [7]Department of Microbiology, Immunology and Transplantation, KU Leuven, Leuven, Belgium. [8]Organ Transplant Center, First Affiliated Hospital, Sun Yat-sen University, Guangzhou, Guangdong, China. [9]Néphrologie-Immunologie Clinique, Hôpital Bretonneau, CHU Tours, Tours, France. [10]Department of Nephrology and Organ Transplantation, Paul Sabatier University, INSERM, Toulouse, France. [11]Department of Nephrology-Dialysis-Transplantation, Centre hospitalier universitaire de Liège, Liège, Belgium. [12]Department of Pathology and Cell Biology, Columbia University Medical Center, New York, NY, USA. [13]Division of Nephrology, Department of Medicine, University of British Columbia, Vancouver, BC, Canada. [14]Kidney Transplant Department, Hospital Clínic i Provincial de Barcelona, Barcelona, Spain. [15]Department of Nephrology, AP-HP Hôpital Henri Mondor, Créteil, Île de France, France. [16]Nephrology Department, Hospital Vall d'Hebrón, Autonomous University of Barcelona, Barcelona, Spain. [17]Department of Pathology, Necker-Enfants Malades Hospital, Assistance Publique - Hôpitaux de Paris, Paris, France. [18]Faculty of Medicine & Dentistry - Laboratory Medicine & Pathology Dept, University of Alberta, Edmonton, AB, Canada. [19]Department of nephrology, arterial hypertension, dialysis and transplantation, University Hospital Centre Zagreb, Zagreb, Croatia. [20]Institute of Pathology, RWTH Aachen University Hospital, Aachen, Germany. [21]Department of Laboratory Medicine and Pathology, Mayo Clinic, Rochester, MN, USA. [22]Department of Renal and Transplantation, University of Adelaide, Royal Adelaide Hospital Campus, Adelaide, SA, Australia. [23]Department of Kidney Transplantation, Necker-Enfants Malades Hospital, Assistance Publique - Hôpitaux de Paris, Paris, France. [24]Renal-Electrolyte and Hypertension Division, Perelman School of Medicine, University of Pennsylvania, Philadephia, PA, USA. [25]These authors contributed equally: Daniel Yoo, Gillian Divard. ✉e-mail: alexandre.loupy@inserm.fr

