## [Peer Review File · Nature Communications]

A Machine Learning-Driven Virtual Biopsy System For Kidney Transplant PatientsREVIEWER COMMENTS

Reviewer #1 (Remarks to the Author):

In this manuscript by Yoo et al, the authors describe their work on applying machine learning in the kidney transplant biopsy. They analyzed 14,032 kidney biopsies at time zero of transplant, using 11 donor parameters to build the virtual biopsy system. These parameters are: donor's age, body mass index, sex, history of hypertension, diabetes, donor cause of death, donor type (living vs deceased), circulatory status at donation, hepatitis C status, serum creatinine, and proteinuria. The investigators aggregated six machine learning models into an ensemble model for predicting arteriosclerosis (cv), arteriolar hyalinosis (ah), interstitial fibrosis and tubular atrophy (IFTA), with multi-AUC of 0.833, 0.773, and 0.830, respectively. The final model demonstrated accurate prediction of the percentage of glomerulosclerosis with a mean absolute error of 5.999.

The manuscript is very well written, and the findings are very well presented. The design of the research is suitable for the goals of the work. The findings are very important and clinically relevant. The study population is extensive and diverse, encompassing 17 transplant centers from various countries.

The promising connection between kidney transplant biopsy findings and easily measured clinical parameters holds significant implications for the selection of kidney transplants, especially those from deceased donors. This breakthrough could potentially result in a substantial reduction in the discard rate of kidneys from deceased donors.

Minor comments:

The parameters employed in the virtual biopsy closely resemble the Kidney Donor Profile Index (KDPI), which has been in use in the USA since late 2014. Given that the study included a population from the USA, could the researchers have utilized the KDPI for the virtual biopsy instead? Furthermore, do the investigators anticipate that the KDPI could potentially substitute the selected parameters in their model, yielding similar outcomes to those of the virtual biopsy?

The term "virtual biopsy" suggests a remote examination of biopsies. However, the research indicates that a physical biopsy may not be necessary, as clinical parameters alone can effectively correlate with biopsy findings. As a result, the name of the model may not accurately describe the nature of the study.

The question arises as to whether the model would remain effective when one or more parameters are missing. The study does not provide explicit clarification on this matter. It is crucial to ascertain the

robustness of the model and its ability to generate reliable predictions when faced with incomplete parameter data. Considering the potential scenarios where certain parameters might be unavailable, further investigation is warranted to evaluate the model's performance. Assessing the sensitivity of the model to missing parameters would enhance our understanding of its practical applicability in real-world settings, where data completeness can vary. Furthermore, addressing the issue of missing parameters would contribute to the model's versatility and enable its potential adoption in cases where complete parameter sets are not always feasible.

One notable observation in the discussion section is the absence of citations referencing similar studies or previous work in the field. While the discussion provides insightful analysis and interpretations of the study's findings, it lacks supporting references to related research.

Reviewer #2 (Remarks to the Author):

The article introduces a machine learning based virtual biopsy system to assess the quality of kidney allografts, based on routinely collected donor parameter. The goal of the system is to replace day-zero biopsies, which are invasive, costly and time consuming.

This multi-center study is generally of high quality and has the potential to lead to a substantial impact in the domain of transplant medicine. Moreover, the availability of an online interface to the virtual biopsy system will facilitate the uptake.

However, I do have a number of concerns regarding the methodology used.

My first concern is the lack of motivation for the chosen machine learning strategy. The authors construct an ensemble method of 6 classifiers (of which several are ensembles themselves). In the end it turns out that this ensemble does not improve over random forests (cfr Table 2), one of the 6 included classifiers. So the final model seems to be overly complex, and the same performance can be achieved with only one component of it. Ideally, the authors would have used their development cohort to compare the results of the 6 classifiers, as well as their ensemble, and then would have picked the best performing one as final model. I understand that the authors have fixed their ML strategy beforehand, but given the no free lunch theorem, it would have been better to optimise the model based on their data, just as they optimise the hyperparameter values. In addition, currently it is also not clear if the complex model is outperforming logistic regression, which could have been included as a (more interpretable, especially given the limited number of features) baseline.

The text states that "hyperparameters were optimized by robust 3-times repeated 10-folds cross-validation when tuning the models". Then, to assess model performance, "For the internal validation, the performance was assessed in 30 resamples from the 3-times repeated 10-folds cross-validation on the derivation cohort". This means that, basically, the same data was used to pick the best hyperparameter values and to report model performance, which may result in overly optimistic results for the internal validation. Ideally, the authors would have used a nested cross-validation set-up or at least a different tuning and test set. Moreover, the authors should specify the hyperparameters that were tuned, the specific values that were included in the tuning process, and the measure that was optimized.

I'm not convinced that the current procedure employed for feature ranking is very useful. By averaging rankings made by different (5 of the 6) individual components of the ensemble, it is hard to get an overall impression of the feature importance. I would advise the authors to run a permutation-based feature importance test (which is probably what is used for several of the individual components) using the complete ensemble.

Finally, although I appreciate the conducted pilot study, it seems to break the flow of the paper. It is not clear what the goal and conclusion of this study are. The work was motivated as a way to avoid the time consuming day-zero biopsies (cold ischemia time). But then in the pilot study, the goal seems to be to check how many discarded kidneys would have been accepted if the virtual biopsy system was used. It seems that the kidneys were discarded without the use of a day-zero biopsy "discarded for quality reasons based on clinical donor parameters and the Kidney Donor Profile Index (KDPI) only". So it is unclear what to conclude from this study, or from the finding that 45% of discarded kidneys would have been accepted unanimously. Would they also have been accepted with day-zero biopsy information? The text is also ambiguous as to whether clinical donor parameters and KDPI were provided to the physicians (line 441 vs line 443-444).

Minor textual comments:

L255: by resampling random values -> by resampling random kidneys (a value is normally interpreted as the value of one predictor)

L525: weird/incorrect sentence starting at this line

Authors' responses to reviewers' comments regarding the manuscript NCOMMS-23-28413 "A Machine Learning-Driven Virtual Biopsy System For Kidney Transplant Patients"

Answers to Reviewer #1

In this manuscript by Yoo et al, the authors describe their work on applying machine learning in the kidney transplant biopsy. They analyzed 14,032 kidney biopsies at time zero of transplant, using 11 donor parameters to build the virtual biopsy system. These parameters are: donor's age, body mass index, sex, history of hypertension, diabetes, donor cause of death, donor type (living vs deceased), circulatory status at donation, hepatitis C status, serum creatinine, and proteinuria. The investigators aggregated six machine learning models into an ensemble model for predicting arteriosclerosis (cv), arteriolar hyalinosis (ah), interstitial fibrosis and tubular atrophy (IFTA), with multi-AUC of 0.833, 0.773, and 0.830, respectively. The final model demonstrated accurate prediction of the percentage of glomerulosclerosis with a mean absolute error of 5.999.

The manuscript is very well written, and the findings are very well presented. The design of the research is suitable for the goals of the work. The findings are very important and clinically relevant. The study population is extensive and diverse, encompassing 17 transplant centers from various countries.

The promising connection between kidney transplant biopsy findings and easily measured clinical parameters holds significant implications for the selection of kidney transplants, especially those from deceased donors. This breakthrough could potentially result in a substantial reduction in the discard rate of kidneys from deceased donors.

We thank the reviewer for the feedback and the detailed comments and suggestions. We also believe that the Virtual Biopsy System has the potential to fill in the gaps of missing day-zero kidney allograft biopsies to avoid potential harmful misinterpretations of the post-transplant biopsy findings for patients. We appreciate the recognition of the international consortium efforts to demonstrate the clinical impact of this companion system in a large number of day-zero biopsies from multi-continental cohorts, to allow a granular assessment of diverse populations. We provide below additional data and clarifications to answer the comments of the reviewer.

Minor comments:

The parameters employed in the virtual biopsy closely resemble the Kidney Donor Profile Index (KDPI), which has been in use in the USA since late 2014. Given that the study included a population from the USA, could the researchers have utilized the KDPI for the virtual biopsy instead? Furthermore, do the investigators anticipate that the KDPI could potentially substitute the selected parameters in their model, yielding similar outcomes to those of the virtual biopsy?

The Kidney Donor Risk Index (KDRI)/Kidney Donor Profile Index (KDPI) and the Virtual Biopsy System have two distinct aims.

1. The KDRI/KDPI was primarily developed for deceased donors only. It is used to assess a kidney offer by predicting relative risk of post-transplant kidney graft failure (with relatively low prediction performances (PMID: 19623019)). Although KDRI/KDPI was not designed to predict histological lesions, it was thereafter used to predict day-zero biopsy findings (it showed poor performances with an AUC of 0.64 (PMID: 29132985)).

2. On the other hand, the Virtual Biopsy System was specifically designed to predict the day-zero kidney graft biopsy results and Banff scores (severity and grades of lesions) in deceased and living donors, and shows good prediction performances. In the end, the ultimate goals of the Virtual Biopsy System were to:

- i) identify the kidneys that need a biopsy,
- ii) accelerate the allocation process,
- iii) decrease cold ischemia time to improve graft outcomes,
- iv) provide a baseline of the graft histology to which the findings of subsequent biopsies of the kidney allograft can be compared.

Following the reviewer's remark, we nevertheless investigated whether KDPI could predict the day-zero biopsy lesions.

To achieve this goal, we developed a new model using solely the KDPI score to predict Banff biopsy scores. We developed four KDPI-based models to predict the four day-zero lesions. We used the imputed datasets that were used for the original Virtual Biopsy System. Biopsies from living donors and with missing ethnicity, height, or weight were excluded. We followed the Organ Procurement and Transplantation Network (OPTN)'s guideline based on the database as of April 07, 2023 (https://optn.transplant.hrsa.gov/media/wnmnxxzu/kdpi_mapping_table.pdf) to calculate KDRI and KDPI. We first calculated the KDRI RAO (PMID: 19623019), then converted it to scaled KDRI with the 2022 scaling factor, 1.33586831546044. Next, we converted it to the KDPI. Up-sampling was performed for the three categorical lesions (Banff cv, ah, and IFTA scores). Ensemble models were developed by aggregating random forest (RF), extreme

gradient boosting tree (XGBoost), linear discriminant analysis (LDA), model averaged neural network (avNNet), and multinomial logistic regression (MNOM). For glomerulosclerosis, LDA and MNOM were excluded because they are exclusively designed to predict categorical variables (classification). Gradient boosting machine was excluded due to the difficulty of deriving a univariate model. With the train set, we performed 3-times repeated 10-folds cross-validation for internal validation. External validation was stratified into Columbia University cohort and Sun Yat-sen University cohort.

For the derivation cohort, 4,241 biopsies were used, and for the external validation cohort, 1,124 biopsies (920 from Columbia University medical center and 204 from Sun Yat-sen University) were used for this analysis. The mean KDPI was 53.43 (SD 29.49) in the derivation cohort; in the external validation cohort, the mean KDPI was 63.24 (SD 26.63).

Table 1 demonstrates the performance of the models with KDPI as a parameter. The new model using the KDPI parameter showed the multi-AUCs of 0.688 (vs 0.833 in the original Virtual Biopsy System), 0.644 (vs 0.773), 0.716 (vs 0.830), for cv, ah, and IFTA lesions during the internal validation, respectively. Predicting the glomerulosclerosis also showed lower performance than the original Virtual Biopsy System (MAE 6.647 vs 5.999), which used the full donor parameters. Most importantly, during the external validations, the simplified KDPI model failed to achieve good performance in predicting cv, ah, and IFTA with 0.625, 0.668, 0.638 for Columbia University cohort, respectively, and 0.659, 0.552, 0.710 for Sun Yat-sen University cohort, respectively. In comparison, our original Virtual Biopsy System showed the good performances with the multi-AUCs of 0.740, 0.733, 0.723, for cv, ah, and IFTA lesions for Columbia University cohort, respectively, and 0.740, 0.736, 0.798 for Sun Yat-sen University cohort, respectively.

In general, other than predicting the glomerulosclerosis, the models with the KDPI model showed poor performance in predicting all three categorical lesion scores reflection arteriosclerosis, arterial hyalinosis, and interstitial inflammation and tubular atrophy (Banff cv, ah, and IFTA scores). This result aligns with the previously mentioned study (PMID: 29132985). The likely reason for this lower performances could be attributed to the simplification of donor parameters into a single score through the KDPI parameter, which might result in a loss of the nuanced information they originally encompassed.

Table 1 | KDPI as a parameter in predicting day-zero kidney biopsy results

Model	Cohort	Validation	Multi-AUC			Mean Absolute Error
			cv	ah	IFTA	Glomerulosclerosis
Virtual Biopsy System	Internal	Cross-validation	0.833	0.773	0.830	5.999
	External	Columbia University	0.740	0.733	0.723	5.200
		Sun Yat-sen University	0.740	0.736	0.798	4.608
KDPI	Internal	Cross-validation	0.688	0.644	0.716	6.647
	External	Columbia University	0.625	0.668	0.638	4.947
		Sun Yat-sen University	0.659	0.552	0.710	4.193

We now added a subsection in Methods and Results to add this sensitivity analysis. The table was added as Supplementary Table 13.

Methods:

“Kidney donor profile index (KDPI)

We conducted a sensitivity analysis to investigate whether KDPI could predict the day-zero biopsy lesions. We developed a model using only the KDPI score. Biopsies from living donors and those with missing ethnicity, height, or weight data were excluded from the imputed dataset. OPTN guidelines, based on the database as of April 07, 2023, were followed for KDPI calculations. An ensemble of RF, XGBoost, LDA, avNNet, and MNOM models was employed. LDA and MNOM were excluded for predicting glomerulosclerosis lesion. GBM was excluded due to the difficulty of deriving a univariate model.”

Results:

“Performance of kidney donor profile index (KDPI) score

The derivation cohort included 4,241 biopsies, and the external validation cohort comprised 1,124 biopsies (920 from Columbia University medical center and 204 from Sun Yat-sen University). The mean KDPI was 53.43 (SD 29.49) in the derivation cohort and 63.24 (SD 26.63) in the external validation cohort.

Supplementary Table 13 shows model performance with KDPI as a parameter. The KDPI-based model achieved multi-AUCs of 0.688, 0.644, and 0.716 for cv, ah, and IFTA lesions during internal validation, respectively. Predicting glomerulosclerosis performed with the MAE of 6.647. During external validations, the KDPI-based model showed predictive performance for cv, ah, and IFTA, achieving multi-AUCs of 0.625, 0.668, and 0.638 for the Columbia University cohort, and 0.659, 0.552, and 0.710 for the Sun Yat-sen University cohort, respectively.”

The term "virtual biopsy" suggests a remote examination of biopsies. However, the research indicates that a physical biopsy may not be necessary, as clinical parameters alone can effectively correlate with biopsy findings. As a result, the name of the model may not accurately describe the nature of the study.

We understand the reviewer's query. Our consortium chose this term, "Virtual Biopsy System", to express the idea that the virtual biopsy is created by computer technology; it therefore exists in the virtual world, and does not exist in the physical world. It is thus our understanding that this term adequately reflects reality. If the reviewer agrees, we would like to keep this terminology and the following remark. We are now better discussing this notion in the revised manuscript.

The following text was added in the introduction.

"The virtual biopsy system, an artificial intelligence (AI) model, provides virtual results that would have been obtained if a biopsy would have been performed."

The question arises as to whether the model would remain effective when one or more parameters are missing. The study does not provide explicit clarification on this matter. It is crucial to ascertain the robustness of the model and its ability to generate reliable predictions when faced with incomplete parameter data. Considering the potential scenarios where certain parameters might be unavailable, further investigation is warranted to evaluate the model's performance. Assessing the sensitivity of the model to missing parameters would enhance our understanding of its practical applicability in real-world settings, where data completeness can vary. Furthermore, addressing the issue of missing parameters would contribute to the model's versatility and enable its potential adoption in cases where complete parameter sets are not always feasible.

We understand the reviewer's remark. Data availability is crucial to ascertain the usability of a prediction model in clinical practice (PMID: 35013569).

However, adapting the Virtual Biopsy System to handle missing parameters is beyond the scope of the present study. Indeed, once a machine learning model is developed, it cannot compute results when needed parameters are missing. One way to handle this issue is to live-imputing the missing parameters with the original complete cohort (PMID: 33482294). Another way is to remove the missing parameters from the development cohorts to redevelop a model, which will learn different patterns from the original full model.

However, none of the methods fully fulfill the loss of information caused by missing parameters.

For these reasons, 11 simple donor parameters were selected as part of the study design because they were easily accessible in most transplant centers worldwide, which is important for the generalizability of the findings. These parameters are: 1) donor's age, 2) body mass index, 3) sex, 4) history of hypertension, 5) diabetes, 6) donor cause of death, 7) donor type (living vs deceased), 8) circulatory status at donation, 9) hepatitis C status as well as renal function parameters including 10) serum creatinine, 11) and proteinuria. These are routinely collected parameters all over the world. It is therefore unlikely that a transplant center lacks these parameters.

We have clarified this point in the study limitation.

“Fourth, additional predictors, such as gene expression or new biomarkers, beyond the 11 donor parameters used to derive the virtual biopsy, may improve its performances. However, the parameters used in this study are the most commonly accessible, and including less standard ones might not only increase the number of missing data but also reduce generalizability by increasing the risk of parameters missing.”

One notable observation in the discussion section is the absence of citations referencing similar studies or previous work in the field. While the discussion provides insightful analysis and interpretations of the study's findings, it lacks supporting references to related research.

We are now referring to the literature search we performed in the discussion section.

Systematic review

We searched PubMed and MEDLINE from January 2000 to January 2022, using the terms “non-invasive”, “biopsy”, “predict”, and “machine learning” without language restrictions. Our search found overall 164 studies from all medical fields. We removed 12 studies predicting a single disease diagnosis (e.g. cancer). 124 studies used histological images and 28 were related to omics-based diagnoses. Overall, in all medical fields, there was no published study on generating a virtual biopsy to assess the presence and severity of biopsy lesions using a combination of non-invasive parameters.

We are now adding a paragraph in the discussion section referencing similar studies.

“Our literature search (Supplementary Method 2) revealed a dearth of studies that address the creation of a virtual biopsy for evaluating biopsy lesion presence and severity by utilizing

non-intrusive factors such as donor parameters. Meanwhile, non-invasive diagnosis using machine learning has been studied. Yin et al. demonstrated that the potential of multiple machine learning classifiers in distinguishing histological features in bladder tumor images. Detecting kidney biopsy results has been explored predominantly with histological images using deep learning. In 2018, Marsh et al. developed a convolutional neural networks (CNN) model to identify and classify glomerulosclerosis in day-zero kidney biopsies, improving pre-transplant evaluation. Hara et al. presented a U-Net based segmentation model for classifying normal and abnormal tubules in kidney biopsies. However, a need persists to compensate for the absence of day-zero biopsy for kidney allografts by virtually assessing the presence and severity of biopsy lesions using non-invasive donor parameters.”

Answers to Reviewer #2

The article introduces a machine learning based virtual biopsy system to assess the quality of kidney allografts, based on routinely collected donor parameter. The goal of the system is to replace day-zero biopsies, which are invasive, costly and time consuming.

This multi-center study is generally of high quality and has the potential to lead to a substantial impact in the domain of transplant medicine. Moreover, the availability of an online interface to the virtual biopsy system will facilitate the uptake.

We would like to thank the reviewer for the comments. We have addressed all remarks as will be seen in the details point-by-point response below.

However, I do have a number of concerns regarding the methodology used.

My first concern is the lack of motivation for the chosen machine learning strategy. The authors construct an ensemble method of 6 classifiers (of which several are ensembles themselves). In the end it turns out that this ensemble does not improve over random forests (cfr Table 2), one of the 6 included classifiers. So the final model seems to be overly complex, and the same performance can be achieved with only one component of it. Ideally, the authors would have used their development cohort to compare the results of the 6 classifiers, as well as their ensemble, and then would have picked the best performing one as final model. I understand that the authors have fixed their ML strategy beforehand, but given the no free lunch theorem, it would have been better to optimise the model based on their data, just as they optimise the hyperparameter values.

We thank the reviewer for this important remark.

Our initial strategy was to aggregate the six popular machine learning algorithms to decrease the bias and maximize the generalizability (Wolpert, David H. "Stacked generalization." Neural networks 1992). We are now providing all machine learning classifiers' performances in both internal and external validations (Table 2). As demonstrated in this Table 2, no single machine learning model performs well in all scenario, except for the ensemble models. Hence, we chose the aggregated ensemble models for the virtual biopsy system, which have shown more stable and robust predictions in both internal and two external validation cohorts.

Table 2 | Machine learning classifiers' and ensemble models' performances in internal and external validation cohorts.

Cohort	Validation	Model	Multi-AUC			Mean Absolute Error
			cv	ah	IFTA	Glomerulosclerosis
Internal	Cross-validation	Random Forest	0.836	0.774	0.830	5.807
		Gradient Boosting Machine	0.807	0.750	0.805	6.486
		Extreme Gradient Boosting Tree	0.830	0.767	0.827	5.768
		Linear Discriminant Analysis	0.761	0.703	0.750	-*
		Model Averaged Neural Network	0.777	0.720	0.757	6.573
		Multinomial Logistic Regression	0.763	0.706	0.753	-*
		Ensemble Model	0.833	0.773	0.830	5.999
External	Columbia University	Random Forest	0.701	0.685	0.683	5.417
		Gradient Boosting Machine	0.723	0.716	0.748	4.989
		Extreme Gradient Boosting Tree	0.687	0.658	0.686	5.095
		Linear Discriminant Analysis	0.754	0.772	0.749	-*
		Model Averaged Neural Network	0.743	0.711	0.62	5.268
		Multinomial Logistic Regression	0.754	0.764	0.731	-*
		Ensemble Model	0.740	0.733	0.723	5.200
	Sun Yat-sen University	Random Forest	0.718	0.678	0.808	4.600
		Gradient Boosting Machine	0.706	0.745	0.817	4.188
		Extreme Gradient Boosting Tree	0.683	0.650	0.738	4.602
		Linear Discriminant Analysis	0.742	0.744	0.816	-*
		Model Averaged Neural Network	0.726	0.670	0.680	4.336
		Multinomial Logistic Regression	0.729	0.741	0.814	-*
		Ensemble Model	0.740	0.736	0.798	4.608

* Linear discriminant analysis and multinomial logistic regression are not developed for regression but for classification. AUC=area under the curve (higher the better). MAE=mean absolute error (lower the better).

We now clarified a Method subsection, *Development of the virtual biopsy system*, for the reasoning of selecting the ensemble model with “no free lunch” theorem.

“Then, we aggregated the classification models by averaging probabilities provided by each model: this generated an ensemble model, or meta-classifier, which is aimed at decreasing bias and overfitting to take into account the “no free lunch” theorem.”

In addition, currently it is also not clear if the complex model is outperforming logistic regression, which could have been included as a (more interpretable, especially given the limited number of features) baseline.

We described in the above Table 2 the results of all machine learning classifiers and ensemble models' performances in internal and two external validation cohorts. The Table 2 shows the logistic regression does not outperform other complex models during the internal validation. With the ensemble method, which aggregates both the complex and non-complex models, the ensemble models overall obtain the benefits of all models (statistical power and generalizability).

To note, generating models to outperform the logistic regression was neither our aim nor our intention. We wanted to take into account the “no free lunch” theorem and avoid using one single optimized model in all situations, whether it is logistic regression or a random forest. The trade-off of “complexity” or “computing time” of the ensemble model is also expected and demonstrated by the “no free lunch” theorem.

The text states that "hyperparameters were optimized by robust 3-times repeated 10-folds cross-validation when tuning the models". Then, to assess model performance, "For the internal validation, the performance was assessed in 30 resamples from the 3-times repeated 10-folds cross-validation on the derivation cohort". This means that, basically, the same data was used to pick the best hyperparameter values and to report model performance, which may result in overly optimistic results for the internal validation. Ideally, the authors would have used a nested cross-validation set-up or at least a different tuning and test set.

We thank the reviewer for this comment, and we are now clarifying this methodological point. With the large derivation cohort of 12,402 biopsy samples from heterogeneous and various data sources, we are confident in performing 3-times repeated 10-folds cross-validation for internal validation. Wainer and Cawley showed comparable results between a flat-procedure and a nested cross-validation (Wainer, Jacques, and Gavin Cawley. "Nested cross-validation when selecting classifiers is overzealous for most practical applications."

Expert Systems with Applications 2021). The authors also stated that “flat-procedure will, on average, perform as well as the one that would be selected by the nested cross-validation procedure, for most practical purposes”.

Moreover, we provided additional layer of model generalizability by performing the validation of the system in various subpopulations and clinical scenarios. Since this partitioning is not equivalent to the cross-validation, we further provided the robustness of the model.

Most importantly, it is our understanding that the only valid way to assess whether the internal validation is overly optimistic is to assess the performance in an external validation. We confirm that our ensemble models show performance in the external validation that was comparable to that of the internal validation.

We now added this point in the limitation in the discussion section.

“Last, other sampling methods such as nested cross-validation may help provide more precise prediction performances. However, with the large derivation cohort from heterogeneous and various data sources, we are confident in performing 3-times repeated 10-folds cross-validation for internal validation. Moreover, we performed model assessments in subpopulations and various clinical scenarios. Finally, we showed the model performances are comparable in internal and external validations.”

Nonetheless, we performed a new model derivation using the split train and test set from the derivation cohort as the reviewer suggested. The split was done with 8:2 stratified random sampling. Hyperparameter optimization was processed during the 3-times repeated 10-folds cross-validation in the train set. Internal validation was performed in both on the cross-validation and on the split test set. After the internal validation, the Virtual Biopsy System was rederived with the full derivation cohort with the optimized hyperparameter. The rederived Virtual Biopsy System was used to perform the external validation in Columbia University and Sun Yat-sen University cohorts. We provide the Table 3 below to demonstrate the overall performance of the newly developed Virtual Biopsy System as the reviewer suggested. The newly developed Virtual Biopsy System shows comparable performance in both internal and external validations compared to the original Virtual Biopsy System. This indicates that the original 3-times repeated 10-folds cross-validation methodology for internal validation provided precise predictive performance and did not yield overly optimistic results.

Table 3 | Machine learning classifiers’ and ensemble models’ performances in internal (cross-validation and test set) and external validation cohorts.

Cohort	Validation	Model	Multi-AUC			Mean Absolute Error
			cv	ah	IFTA	Glomerulosclerosis
Internal	Cross-validation	Random Forest	0.814	0.753	0.809	6.000
		Gradient Boosting Machine	0.800	0.730	0.791	6.360
		Extreme Gradient Boosting Tree	0.809	0.749	0.805	5.872
		Linear Discriminant Analysis	0.764	0.699	0.751	-*
		Model Averaged Neural Network	0.771	0.708	0.766	6.557
		Multinomial Logistic Regression	0.766	0.701	0.752	-*
		Ensemble Model	0.820	0.757	0.814	5.927
	Test set	Random Forest	0.832	0.770	0.824	6.025
		Gradient Boosting Machine	0.802	0.751	0.801	6.442
		Extreme Gradient Boosting Tree	0.823	0.757	0.834	5.890
		Linear Discriminant Analysis	0.752	0.718	0.730	-*
		Model Averaged Neural Network	0.758	0.720	0.764	6.752
		Multinomial Logistic Regression	0.754	0.723	0.736	-*
		Ensemble Model	0.825	0.783	0.821	5.855
External	Columbia University	Random Forest	0.701	0.685	0.711	5.369
		Gradient Boosting Machine	0.718	0.678	0.717	5.955
		Extreme Gradient Boosting Tree	0.701	0.680	0.674	4.963
		Linear Discriminant Analysis	0.754	0.772	0.749	-*
		Model Averaged Neural Network	0.753	0.746	0.719	5.144
		Multinomial Logistic Regression	0.754	0.764	0.731	-*
		Ensemble Model	0.742	0.734	0.738	5.076
	Sun Yat-sen University	Random Forest	0.718	0.678	0.833	4.582
		Gradient Boosting Machine	0.698	0.663	0.766	4.708
		Extreme Gradient Boosting Tree	0.670	0.708	0.763	4.470
		Linear Discriminant Analysis	0.742	0.744	0.816	-*
		Model Averaged Neural Network	0.741	0.735	0.856	4.342
		Multinomial Logistic Regression	0.729	0.741	0.814	-*
		Ensemble Model	0.736	0.740	0.800	4.497

* Linear discriminant analysis and multinomial logistic regression are not developed for regression but for classification. AUC=area under the curve (higher the better). MAE=mean absolute error (lower the better).

Moreover, the authors should specify the hyperparameters that were tuned, the specific values that were included in the tuning process, and the measure that was optimized.

As suggested by the reviewer, we now provide the hyperparameters that were tuned, the chosen final hyperparameters, and the measure that was optimized (Supplementary Table 5). The main manuscript and the supplementary appendix have been revised accordingly.

Supplementary Table 5 | Hyperparameters tuning and results

Hyperparameters of the machine learning models were tuned during 3-times repeated 10-folds cross-validation. For three ordinal Banff lesion scores (arteriosclerosis [Banff cv score], arteriolar hyalinosis [Banff ah score], interstitial fibrosis and tubular atrophy [Banff IFTA score]), the Hand and Till's multi-AUC (higher the better) was measured to optimize. For the continuous Banff lesion, glomerulosclerosis (percentage of sclerotic glomeruli), the mean absolute error (MAE, lower the better) was measured to optimize. For cv, ah, and IFTA scores, averaging all machine learning models generated an ensemble model. For glomerulosclerosis, a linear regression model of regression models to create an ensemble model.

Supplementary Table 5.1 | Arteriosclerosis (Banff cv score)

Machine learning models	Hyperparameters
Random Forest	mtry=4
Gradient Boosting Machine	n.trees=700 interaction.depth=13 shrinkage=0.01 n.minobsinnode=7
Extreme Gradient Boosting Tree	nrounds=54 max_depth=18 eta=0.1852479 gamma=0.02767602 colsample_bytree=0.6063756 min_child_weight=0.9 subsample=0.790576
Linear Discriminant Analysis	-
Model Averaged Neural Network	size=25 decay=0.1 bag=TRUE
Multinomial Logistic Regression	decay=0.001

Supplementary Table 5.2 | Arteriolar hyalinosis (Banff ah score)

Machine learning models	Hyperparameters
Random Forest	mtry=4
Gradient Boosting Machine	n.trees=700 interaction.depth=13 shrinkage=0.01 n.minobsinnode=5
Extreme Gradient Boosting Tree	nrounds=27 max_depth=18 eta=0.06210775 gamma=0.01385926 colsample_bytree=0.8300242 min_child_weight=1.1 subsample=0.8261786
Linear Discriminant Analysis	-
Model Averaged Neural Network	size=15 decay=0.1 bag=TRUE
Multinomial Logistic Regression	decay=0.001

Supplementary Table 5.3 | Interstitial fibrosis and tubular atrophy (Banff IFTA score)

Machine learning models	Hyperparameters
Random Forest	mtry=4
Gradient Boosting Machine	n.trees=700 interaction.depth=13 shrinkage=0.01 n.minobsinnode=7
Extreme Gradient Boosting Tree	nrounds=38 max_depth=15 eta=0.1508891 gamma=0.04430697 colsample_bytree=0.5812269 min_child_weight=1.9 subsample=0.9993576
Linear Discriminant Analysis	-
Model Averaged Neural Network	size=10 decay=0.01 bag=FALSE
Multinomial Logistic Regression	decay=0.01

Supplementary Table 5.4 | Glomerulosclerosis (percentage of sclerotic glomeruli)

Machine learning models	Hyperparameters
Random Forest	mtry=8
Gradient Boosting Machine	n.trees=700 interaction.depth=13 shrinkage=0.01 n.minobsinnode=5
Extreme Gradient Boosting Tree	nrounds=283 max_depth=18 eta=0.01032906 gamma=0.04123139 colsample_bytree=0.5279746 min_child_weight=0.7 subsample=0.6965341
Model Averaged Neural Network	size=10 decay=0.01 bag=FALSE
Ensemble model (linear regression)	-

I'm not convinced that the current procedure employed for feature ranking is very useful. By averaging rankings made by different (5 of the 6) individual components of the ensemble, it is hard to get an overall impression of the feature importance. I would advise the authors to run a permutation-based feature importance test (which is probably what is used for several of the individual components) using the complete ensemble.

We now enhanced our feature importance analysis as advised by the reviewer by including the previously missing algorithm (linear discriminant analysis [LDA]) and permutation-based feature importance tests. Moreover, we performed the full spectrum of continuous importance rather than only reporting the rankings. Since the ensemble models are averaging scores from the six base machine learning models, we averaged the importance of the 6 models. We revised the manuscript and the figure accordingly.

“Furthermore, to assess the donor parameters that drive the performance of the models, we averaged the feature importance by RF, GBM, XGBoost, LDA (for classification models only), avNNet, and MNOM (for classification models only).”

Figure 1 | Clinical and biological parameters' importance

We performed random forest, gradient boosting machine, extreme gradient boosting tree, linear discriminant analysis, model averaged neural network, and multinomial logistic regression to measure the parameter importance for predicting the day-zero biopsy histological lesion scores during the derivation process. The importance was then averaged for the ensemble model. (a) Donor parameter importance for arteriosclerosis (cv Banff score). (b) Donor parameter importance for arteriolar hyalinosis (ah Banff score). (c) Donor parameter importance for interstitial fibrosis and tubular atrophy (IFTA Banff score). (d) Donor parameter importance for the percentage of sclerotic glomeruli (glomerulosclerosis score).

Finally, although I appreciate the conducted pilot study, it seems to break the flow of the paper. It is not clear what the goal and conclusion of this study are. The work was motivated as a way to avoid the time consuming day-zero biopsies (cold ischemia time). But then in the pilot study, the goal seems to be to check how many discarded kidneys would have been accepted if the virtual biopsy system was used. It seems that the kidneys were discarded without the use of a day-zero biopsy "discarded for quality reasons based on clinical donor parameters and the Kidney Donor Profile Index (KDPI) only". So it is unclear what to conclude from this study, or from the finding that 45% of discarded kidneys would have been accepted unanimously. Would they also have been accepted with day-zero biopsy information? The text is also ambiguous as to whether clinical donor parameters and KDPI were provided to the physicians (line 441 vs line 443-444).

We thank the reviewer for bringing up this point. We agree with the reviewer's comment and removed the pilot study section. The main manuscript and the supplementary appendix have been revised accordingly.

Minor textual comments:

L255: by resampling random values -> by resampling random kidneys (a value is normally interpreted as the value of one predictor)

We thank the reviewer for finding this typo. We corrected this sentence.
“by resampling random kidneys from the severe/higher grades.”

L525: weird/incorrect sentence starting at this line

We corrected this sentence. “Besides, our data collection procedure followed high-quality structured protocols to ensure compatibility across study centers.”

REVIEWERS' COMMENTS

Reviewer #1 (Remarks to the Author):

That authors addressed all the comments, concerns and questions that are raises by the reviewers. No other comments.

Reviewer #2 (Remarks to the Author):

The authors carefully addressed all my comments.

Authors' responses to reviewers' comments regarding the manuscript NCOMMS-23-28413 "A Machine Learning-Driven Virtual Biopsy System For Kidney Transplant Patients"

Answers to Reviewer #1

That authors addressed all the comments, concerns and questions that are raises by the reviewers. No other comments.

We thank the reviewer for the feedback and the detailed comments and suggestions.

Answers to Reviewer #2

The authors carefully addressed all my comments.

We thank the reviewer for the feedback and the detailed comments and suggestions.